# Efficacy and safety of three-dimensional magnetically assisted capsule endoscopy for upper gastrointestinal and small bowel examination

**Dong Jun Oh**[1,☯], **Yea Je Lee**[1,☯], **Sang Hoon Kim**[1], **Joowon Chung**[2], **Hyun Seok Lee**[3], **Ji Hyung Nam**[1], **Yun Jeong Lim**[1] *

1 Department of Internal Medicine, Dongguk University Ilsan Hospital, Goyang, Republic of Korea,
2 Department of Internal Medicine, Nowon Eulji Medical Center, Seoul, Republic of Korea, 3 Department of Internal Medicine, Kyungpook National University Hospital, Daegu, Republic of Korea

☯ These authors contributed equally to this work.
* drlimyj@gmail.com

**Data Availability Statement:** However, we regret to inform you that, due to the involvement of technology transfer and the development of

## Abstract

### Background

Magnetically assisted capsule endoscopy (MACE) showed the feasibility for upper gastrointestinal examination. To further enhance the performance of conventional MACE, it is necessary to provide quality-improved and three-dimensional images. The aim of this clinical study was to determine the efficacy and safety of novel three-dimensional MACE (3D MACE) for upper gastrointestinal and small bowel examination at once.

### Methods

This was a prospective, single-center, non-randomized, and sequential examination study (KCT0007114) at Dongguk University Ilsan Hospital. Adult patients who visited for upper endoscopy were included. The study protocol was conducted in two stages. First, upper gastrointestinal examination was performed using 3D MACE, and a continuous small bowel examination was performed by conventional method of capsule endoscopy. Two hours later, an upper endoscopy was performed for comparison with 3D MACE examination. The primary outcome was confirmation of major gastric structures (esophagogastric junction, cardia/fundus, body, angle, antrum, and pylorus). Secondary outcomes were confirmation of esophagus and duodenal bulb, accuracy for gastric lesions, completion of small bowel examination, 3D image reconstruction of gastric lesion, and safety.

### Results

Fifty-five patients were finally enrolled. The examination time of 3D MACE was 14.84 ± 3.02 minutes and upper endoscopy was 5.22 ± 2.39 minutes. The confirmation rate of the six major gastric structures was 98.6% in 3D MACE and 100% in upper endoscopy. Gastric lesions were identified in 43 patients during 3D MACE, and 40 patients during upper

medical devices in our study, we currently face constraints in sharing the complete raw data. In relation to technology transfer, we would like to inform you that the rights to the data belong to a third party. This situation has introduced legal restrictions that limit our ability to freely share the complete raw data. To access the datasets from our study, please contact: (1) Company representative (Kang, Sangjoon): "kangsj@intromedic.com" for data rights and technology transfer details. (2) Principal investigator (Lim, Yun Jeong) : "drlimyj@gmail. com" for research questions and methodology. B. If endoscopists experienced in both capsule endoscopy and upper endoscopy have adequately practiced with simulator model and animal model, as detailed in "PLoS One. 2021 Oct 5;16(10): e0256519," we are confident that they can fully replicate the findings of our study by following the protocols outlined in our Methods section. C. The authors did not have any special access privileges or advantages that others would not have.

**Funding:** This study was supported by a grant (grant Number: HI19C0665) from the Korean Health Technology R & D project through the Korean Health Industry Development Institute (KHIDI). And the fund holder is the principal investigator (Yun Jeong Lim) of this study. The fund holder had a role in study design, data collection and analysis, decision to publish, and preparation of the manuscript. However, there is absolutely no financial interests between the fund holder and the company.

**Competing interests:** The authors have declared that no competing interests exist.

endoscopy (Sensitivity 0.97). 3D reconstructed images were acquired for all lesions inspected by 3D MACE. The continuous small bowel examination by 3D MACE was completed in 94.5%. 3D MACE showed better overall satisfaction (3D MACE 9.55 ± 0.79 and upper endoscopy 7.75 ± 2.34, p<0.0001). There were no aspiration or significant adverse event or capsule retention in the 3D MACE examination.

## Conclusions

Novel 3D MACE system is more advanced diagnostic modality than the conventional MACE. And it is possible to perform serial upper gastrointestinal and small bowel examination as a non-invasive and one-step test. It would be also served as a bridge to pan-endoscopy.

## Introduction

Capsule endoscopy is a relatively safe and easy-to-apply procedure that allows direct visualization of gastrointestinal (GI) mucosa [1]. However, capsule endoscopy has a disadvantage in that it cannot actively move. Conventional capsule endoscopy can only move with gut peristalsis and gravity [2]. In 2010, a clinical study on external magnetic control of capsule endoscopy in the esophagus and stomach was conducted for the first time [3]. Afterwards, esophageal and/or gastric examinations were performed using various types of magnetic controllers [4]. Among them, a magnetically assisted capsule endoscopy (MACE) with a hand-held controller (MiroCam Navi system, Intromedic Co. Ltd., Seoul, Korea) showed the feasibility for upper GI examination. In addition, it is simple and portable equipment [5]. In addition, several studies have shown that conventional MACE is as accurate as upper endoscopy in identifying Barrett's esophagus, focal gastric or small bowel lesions, and upper GI landmarks with less discomfort [5–8].

However, to further improve the performance of MACE, it requires advanced software to improve the image quality and provide more three-dimensional (3D) views of lesions and structures [5, 8]. Also, more user-friendly and easy-to-manipulate magnetic controller is needed. Recently, a study on small bowel examination using 3D capsule endoscopy was reported. Unlike conventional capsule endoscopy, 3D capsule endoscopy was equipped with dual-stereo camera, enabling 3D image reconstruction and image quality improvement [9]. In addition, a new 3D MACE system developed by combining 3D capsule endoscopy and conventional MACE was successfully performed in porcine gastric examination [10].

We decided to apply this novel 3D MACE with a user-friendly magnetic controller in clinical setting. So, the aim of this clinical study was to confirm the efficacy and safety of novel 3D MACE system for upper GI examination. Also, we assessed the ability of 3D MACE to simultaneously examine the small bowel.

## Materials and methods

### Study design and populations

This was a prospective, single-center, and pilot study conducted at Dongguk University Ilsan Hospital from April 2022 and July 2022. It enrolled adult patients between ages of 19 and 80 who visited the hospital for upper endoscopy due to upper GI symptoms such as dyspepsia and epigastric soreness. Exclusions included patients with dysphagia, GI perforation and bleeding, known Crohn's disease, previous abdominal surgery, pregnancy, and any medical devices that might affect magnetic fields such as intracardiac pacemaker, defibrillator.

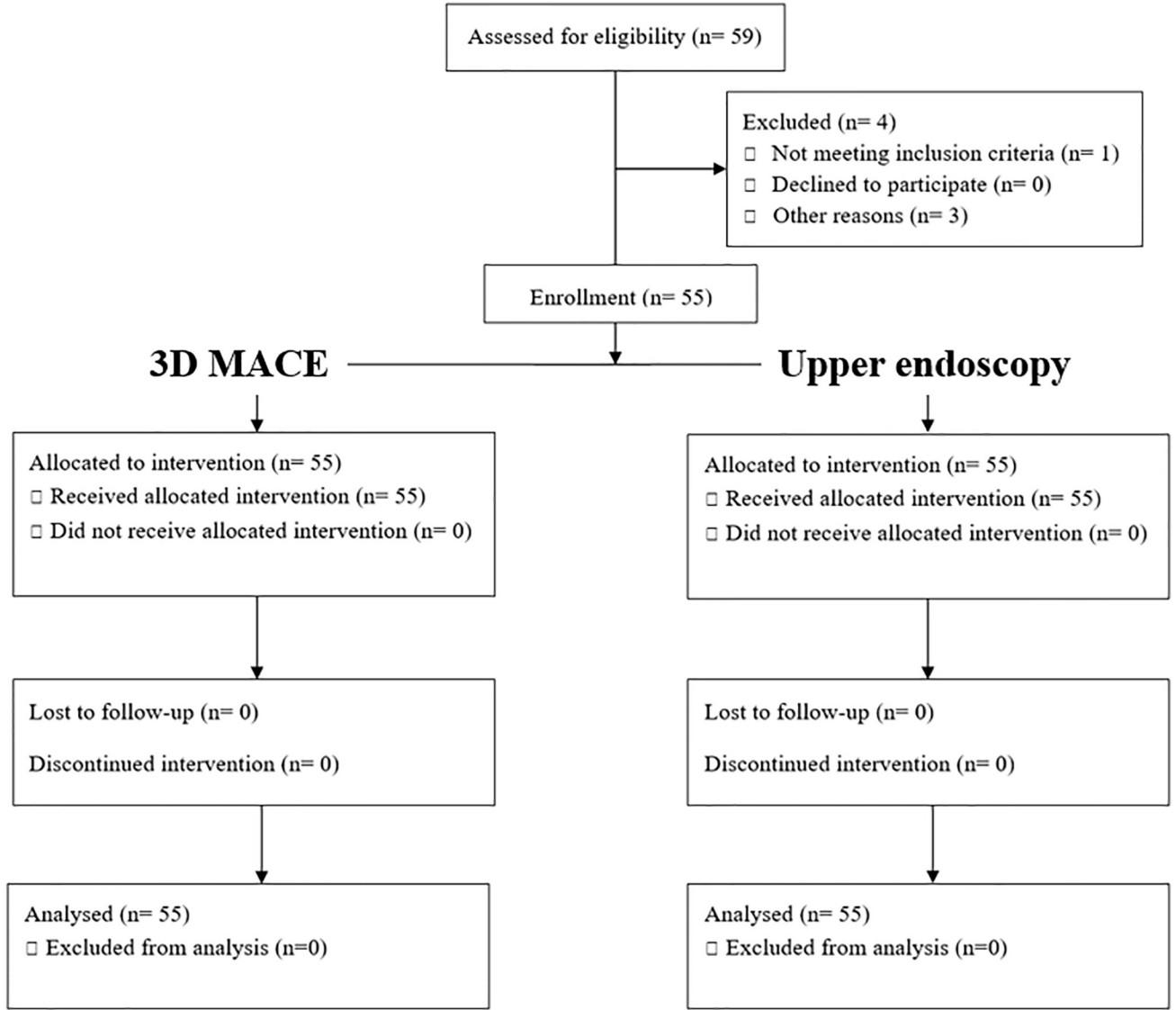

**Fig 1. CONSORT flow chart of study populations.** In this study, upper gastrointestinal examination was performed using two different methods, 3D magnetically assisted capsule endoscopy and upper endoscopy, in the same patient.

All patients provided written informed consent. All patients were anonymized and assigned a code number. They were examined according to the Declaration of Helsinki guidelines. This study was approved by the Institutional Review Board of Dongguk University Ilsan Hospital (IRB No. DUIH 2022-01-038) and registered in the Clinical Research Information Service (CRIS KCT0007114). A total of 59 patients were sequentially enrolled. Of these, three were excluded and a total of 55 patients were finally enrolled. (Fig 1).

### 3D MACE system

A novel 3D MACE system (Fig 2) consisted of a 3D MACE (MiroCam MC 4000-M, Intromedic Co. Ltd, Seoul, Korea), a kettlebell-shaped hand-held magnetic controller, and a data sensor belt with a real-time receiver (MR2000, Intromedic Co. Ltd, Korea) [10].

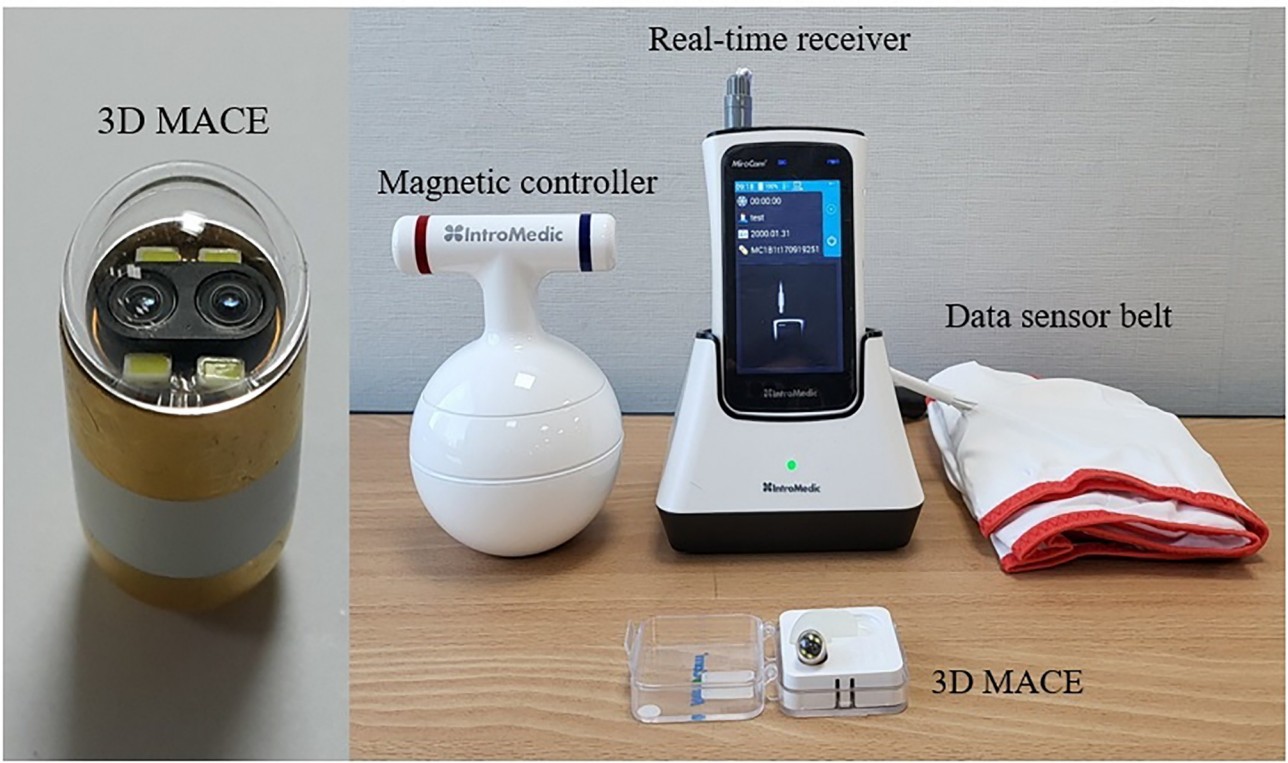

**Fig 2. Novel three-dimension magnetically assisted capsule endoscopy (3D MACE) system.** A novel 3D MACE system consisted of a 3D MACE, a portable hand-held magnetic controller, and a data sensor belt with real-time receiver. 3D MACE was equipped with dual cameras in capsule endoscopy.

**(1) 3D MACE.** The 3D MACE was equipped with dual stereo cameras to enable 3D image reconstruction. This is a compact size of 11 x 25.5 mm. It weighed only 4.5 g, similar to the conventional MACE. It also exhibited the same image resolution (320 x 320 pixels), frame rate (2 frames per second), visual field (170 degrees), and battery time (10 hours) as a conventional MACE [5].

**(2) New hand-held magnetic controller.** We employed a new kettlebell-shaped magnetic controller, distinct from the conventional hammer-shaped and circular stacked controllers [5, 10]. The new controller has the advantage of being easy to hold and move due to its shape. In addition, the N and S poles are marked on the handle of controller, which can help predict the direction of camera in 3D MACE. (S1 Fig)

**(3) Data sensor belt and real-time receiver.** The data sensor belt was attached to the real-time receiver after being wrapped around the patient's abdomen. The real-time receiver was connected to a laptop to display images on a larger screen. The data sensor belt and real-time receiver utilized in this study were of the same model used for conventional small bowel capsule endoscopy (MiroCam MC2000, Intromedic Co. Ltd., Seoul, Korea) [10, 11].

**(4) Software reading viewer.** Images of upper GI and small bowel examination using the 3D MACE system were reviewed using a software-based reading viewer (MiroView 4.0, Intromedic Co. Ltd, Korea). Images could also be reconstructed into 3D images using this reading viewer. Information relevant to this study has been described previously published article [10, 11].

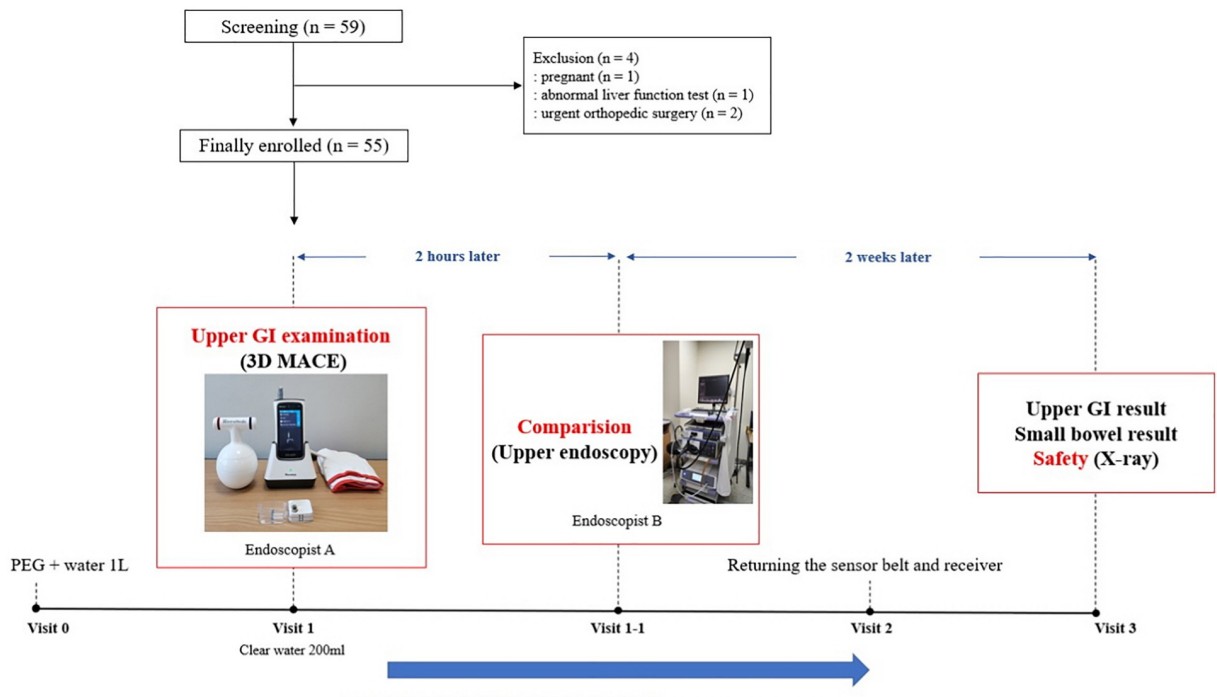

**Fig 3. Schematic of the study protocol.** Three-dimension magnetically assisted capsule endoscopy (3D MACE) was performed for upper gastrointestinal (GI) and small bowel examination at the same time. Two hours later, an upper endoscopy was performed for comparison with 3D MACE examination. Capsule retention and any adverse events were evaluated after two weeks.

## Study protocol

This study was conducted as a sequential (two-staged) upper GI examination (Fig 3) by performing 3D MACE and comparing by upper endoscopy in the same patient. 3D MACE examination was performed first, and upper endoscopy was performed 2 hours later. 3D MACE was manipulated by Oh DJ and upper endoscopy was manipulated by Lim YJ. Both Oh DJ and Lim YJ are expert endoscopists. The 3D MACE system was utilized to simultaneously examine the upper GI and small bowel.

**(1) 3D MACE for upper GI and small bowel examination.** Patients underwent fasting for at least 12 hours with only clear liquids before examination. One hour prior to the examination, each patient consumed 500 ml of a polyethylene glycol (PEG)-based purgative (Coolprep, TaeJoon Pharm Co. Ltd, Seoul, Korea) and 500 ml of water. Before the examination, a data sensor belt was placed over the patient's upper abdomen in the endoscopy room. The patient then ingested 200 ml of clean water and 40 mg of simethicone (TaeJoon Pharm Co. Ltd, Seoul, Korea) to eliminate air bubbles in the stomach. The patient was placed in a left lateral position with a 45-degree head elevation on a mobile stretchers bed (Stryker Corporation, Michigan, USA) that can be adjusted the height and angle. The hand-held magnetic controller was positioned between the patient's scapula, ensuring that the 3D MACE reached the esophagus and came to a halt. Then, the patient swallowed 3D MACE with a cup of water. When the 3D MACE stopped at the esophagus, the controller was moved up and down to observe the esophagus. After esophageal examination, the gastric examination from the esophagogastric (EG) junction to the pylorus was performed according to the previously established manipulation protocol [5, 12]. Once the 3D MACE landed on the stomach, the cardia and fundus were

observed by rotating and tilting the controller in the left lateral position. Then, the patient was placed upright, and the controller was placed in the left upper quadrant and left flank area to observe the fundus and body. After that, the operator moved the controller to right upper quadrant and right flank area to observe the antrum and pylorus. If necessary, the patient was changed to the right lateral position and the antrum was observed. (S2 Fig).

The gastric examination was completed when all gastric structures and abnormal gastric lesions were identified. And 3D MACE was inserted into the duodenum by the controller. If the 3D MACE had trouble passing the pylorus, the examination was ended at the judgment of the endoscopist. After the examination, the patient filled out a questionnaire to rate their satisfaction and discomfort with the 3D MACE. During this time, the 3D MACE was performed continuously for small bowel examination by conventional method of capsule endoscopy.

**(2) Upper endoscopy for upper GI examination.** Two hours after the 3D MACE examination, another endoscopist (Lim YJ) performed an upper GI examination with upper endoscopy (EVIS LUCERA ELITE GIF-HQ290EC, Olympus Co. Ltd., Japan). During upper endoscopy, endoscopist confirmed the all gastric structures, abnormal gastric lesions, and any possible adverse events caused by the 3D MACE. Upper endoscopy was performed as a sedative procedure using midazolam or propofol. When upper endoscopy was over, the patient was managed in the recovery room. After full recovery from sedation, the patient recorded the satisfaction and discomfort related to the upper endoscopy through a separately prepared questionnaire.

**(3) Reading of 3D MACE examination.** After 3D MACE and upper endoscopy examinations, the patient went home with a data sensor belt and real-time receiver still attached for 24 hours. The following day, the patient returned the equipment to the hospital. And, scanned images during 3D MACE examination were uploaded to a reading viewer. The endoscopist reviewed the uploaded images of upper GI structure and abnormal gastric lesions captured during the 3D MACE examination. The endoscopist then assessed images of the small bowel to confirm the presence of small bowel lesions and completion of the examination.

## Outcomes and safety

The primary outcome was to confirm the of major gastric structures with 3D MACE system. A total of 6 major structures were defined as EG junction, cardia/fundus, body, angle, antrum, and pylorus. Mucosal visualization for each structure was also assessed. Failure to structure confirmation was defined as a case where the structure was not visible during examination or was not clearly photographed. The secondary outcomes measured the confirmation of esophagus and duodenal bulb, the accuracy in detecting major gastric lesions (e.g. ulcer, polyp $\geq 1$ cm, cancer) and minor gastric lesions (e.g. inflamed lesion such as erosion, hematin, and hyperemia, polyp $< 1$ cm, other abnormalities), completion rate of small bowel examination in 3D MACE, 3D reconstruction of gastric lesions, and patient satisfaction and adverse events during both 3D MACE.

Patient's satisfaction and adverse events were assessed immediately after 3D MACE and upper endoscopy respectively. Two weeks later, capsule retention and adverse events were also evaluated. The patient was instructed by the researcher to observe and document excretion of the 3D MACE. If it was not excreted or confirmed after 2 weeks, an abdomen radiograph was performed to confirm capsule retention.

## Sample size and statistical analysis

In the test to confirm the gastric structures using conventional MACE, the detection rate of cardia, which has the lowest detection success rate, was 88% [5]. For reliable gastric

examination, the detection rate of gastric structures was considered to be 98%. To ensure the power and stability of the sample size in our study, we conducted a Power Analysis, setting the level of significance (α) at 0.05 and the desired power of the test (β) at 80%. This involved considering a standard deviation of 4.0 based on the successful visualization of gastric structures from the previous study [5], and an effect size of 0.8 in the t-test for the Power Analysis. The ratio of case to control was 1, but 3D MACE and upper endoscopy were performed on the same patient because of the sequence examination and self-controlled design. The number of patients was determined to be 55, considering the dropout rate of 10%.

Statistics on the endoscopist's assessment of upper GI structure and gastric lesions showed the mean, standard deviation, median, minimum, and maximum values. Continuous variables are described by number of patients, mean, standard deviation, median, minimum, and maximum values. Categorical variables are shown as frequencies and percentages. Two sample t-test (or Wilcoxon's rank sum test) and Chi-square test were used to analyze variables. Kappa (κ), sensitivity, and specificity were used to evaluate the agreement and assess each part of upper GI anatomy and gastric lesions between two groups. Statistical significance was set at $p < 0.05$ in both univariate analyses. All statistical analyses were carried out with IBM SPSS Statistics v25.

## Results

### Baseline characteristics

The mean age of enrolled patients was 42.04 ± 10.36 years (range 20 to 69 years). Upper endoscopy was performed at two hours after 3D MACE examination. Midazolam (54 cases) and propofol (1 case of paradoxical reaction to midazolam) were intravenously administered as a sedative upper endoscopy. The gender ratio was almost equal. (S1 Table).

### Upper GI examination by 3D MACE and upper endoscopy

**(1) Confirmation of major gastric structures, esophagus, and duodenal bulb.**   The mean examination time of 3D MACE was 14.84 ± 3.02 minutes (range 6 to 22 minutes) and upper endoscopy was 5.22 ± 2.39 minutes (range 3 to 9 minutes). The 3D MACE identified major gastric structures in 98.6% of all cases, while upper endoscopy in 100% of all cases. Both endoscopic methods confirmed all five gastric structures (cardia/fundus, body, angle, antrum, and pylorus). However, 3D MACE failed to identify the EG junction in one case. The 3D MACE identified major upper GI structures (Fig 4) containing all of the main gastric structures, esophagus, and duodenum in 99.1% of all cases, while upper endoscopy identified major upper GI structures in 100% of all cases. The 3D MACE failed to identify the esophagus in three cases and the EG junction in one case, with a sensitivity of 94.5% for esophagus and 98.2% for EG junction (Table 1).

**(2) Abnormal gastric lesion detection and 3D reconstruction.**   Abnormal gastric lesions (S3 Fig) were identified in 43 of 55 patients who underwent 3D MACE examination. In one case, a gastric ulcer in the healed stage was observed. Most cases consisted of minor lesions with 40 cases (72.2%) gastric inflamed lesions and 2 cases (3.6%) gastric polyps less than 1 cm. After examination, 3D reconstructed images were obtained using a reading viewer for all lesions observed in the 3D MACE group (Fig 5). Abnormal gastric lesions were identified in 40 of 55 patients who underwent upper endoscopy. The gastric ulcer found in 3D MACE was also observed in the same location and shape in upper endoscopy. Remaining cases had minor lesions including 35 cases (63.6%) gastric inflamed lesions, 1 case (1.8%) gastric polyp, and 1 case (1.8%) gastric subepithelial lesion. When 3D MACE was performed, small inflamed lesions and small gastric polyp were additionally found in five cases and one case, respectively. However, gastric subepithelial lesion was not observed in 3D MACE. (Table 2). In the case of

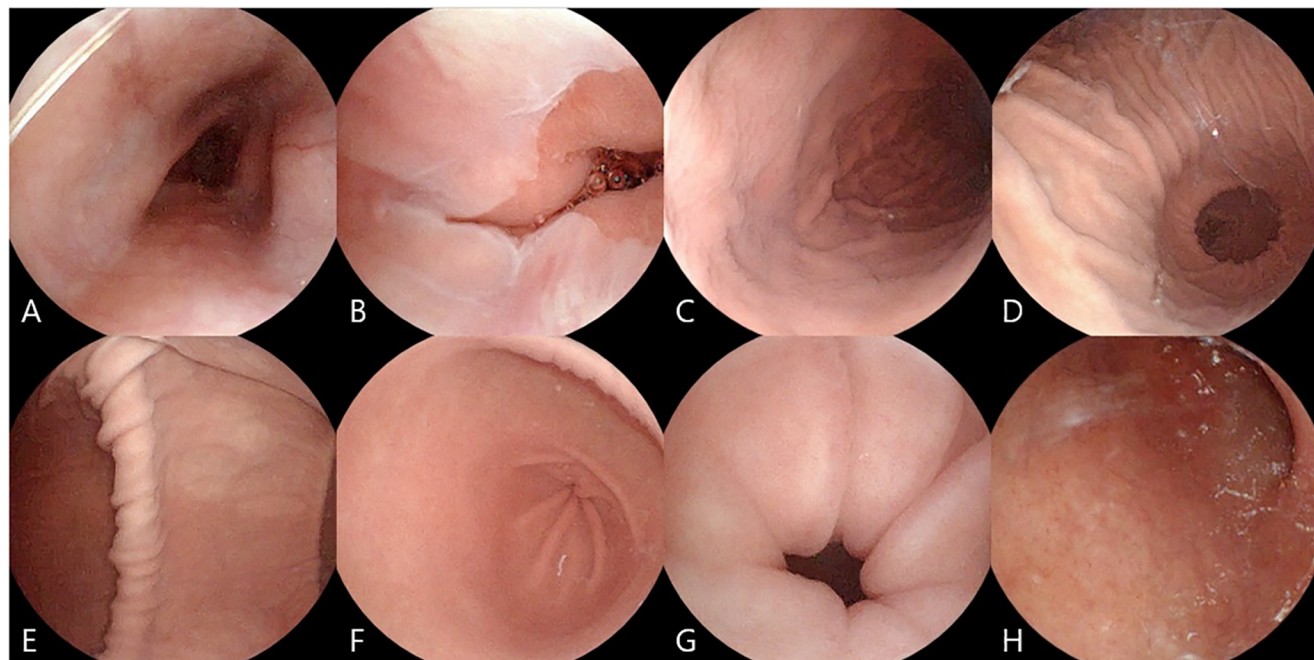

**Fig 4. Eight major upper gastrointestinal structures identified by three-dimensional magnetically assisted capsule endoscopy (3D MACE) examination.** [A] Esophagus, [B] Esophagogastric junction, [C] Cardia and fundus, [D] Body, [E] Angle, [F] Antrum, [G] Pylorus, and [H] Duodenal bulb.

missed lesions, the authors reviewed the 3D MACE images and concluded that the lesions were insignificant. The endoscopist explained that the patients needed periodic follow-up endoscopy.

**(3) Mucosal visualization of upper GI structures during 3D MACE.** Mucosal visualization for each structure was also assessed. Less than 70% of the visible part of the mucosa was defined as poor, 70–90% as fair, and more than 90% as good [13]. Of the eight upper GI structures in a total of 55 cases, 81.8% (360/440) had good visualization, 15.7% (69/440) had fair

**Table 1. Confirmation of upper gastrointestinal (GI) structures using the three dimensional magnetically assisted capsule endoscopy (3D MACE) and upper endoscopy.**

|  | 3D MACE (n = 55) | Upper endoscopy (n = 55) | Remarks |
|---|---|---|---|
| Anatomic structure |  |  |  |
| esophagus | 94.5% | 100% | 3 signal noises |
| EG[$] junction | 98.2% | 100% | 1 signal noise |
| cardia & fundus | 100% | 100% |  |
| body | 100% | 100% |  |
| angle | 100% | 100% |  |
| antrum | 100% | 100% |  |
| pylorus | 100% | 100% |  |
| duodenal bulb | 100% | 100% |  |
| Confirmation rate |  |  |  |
| gastric anatomies | 98.6% | 100% | $p = 0.3195$ |
| upper GI anatomies | 99.1% | 100% | $p = 0.1001$ |

[a]EG; esophagogastric

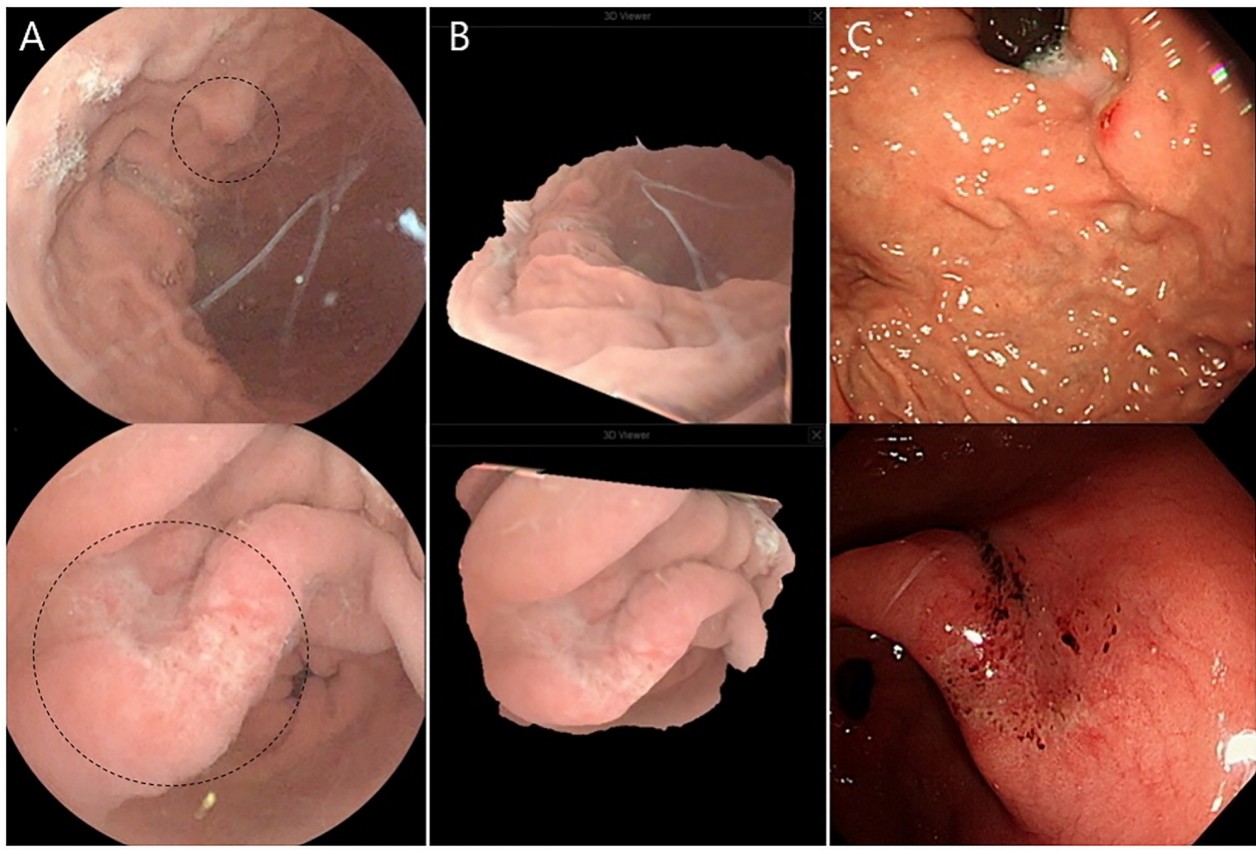

**Fig 5. Abnormal gastric lesion detection and three-dimensional (3D) image reconstruction using a 3D magnetically assisted capsule endoscopy (MACE) system.** [A] Elevated erosion (top image) and ulceration (bottom image) were inspected during 3D MACE examination. Image brightness and definition was enhanced using a reading viewer. [B] Images of the gastric lesions were 3D reconstructed and rotated using a reading viewer. [C] Erosion and ulceration also were observed by an upper endoscopy in same location and shape. Images of lesion obtained by 3D MACE and upper endoscopy were almost identical.

**Table 2. Abnormal gastric lesion detection using the 3 dimensional magnetically assisted capsule endoscopy (3D MACE) and upper endoscopy.**

| Impression | 3D MACE (n = 55) | Upper endoscopy (n = 55) | Remarks |
|---|---|---|---|
| Major lesions | | | |
| ulcer | 1 (1.8%) | 1 (1.8%) | |
| polyp ≥ 1cm | 0 (0.0%) | 0 (0.0%) | |
| cancer | 0 (0.0%) | 0 (0.0%) | |
| Minor lesions | | | Sensitivity 0.97 |
| inflamed lesion | 40 (72.7%) | 37 (67.3%) | |
| focal | 9 (16.3%) | 5 (9.1%) | |
| diffuse | 31 (56.4%) | 32 (58.2%) | |
| polyp < 1cm | 2 (3.6%) | 1 (1.8%) | |
| subepithelial lesion | 0 (0.0%) | 1 (1.8%) | |

**Table 3. Continuous small bowel examination using three dimensional magnetically assisted capsule endoscopy (3D MACE).**

| Variables | Patients (n = 55) |
|---|---|
| Small bowel transit time, mins | 260.34 (± 101.14) |
| Complete examination | 52 (94.5%) |
| Capsule retention | 0 (0%) |
| Small bowel lesions | |
| no | 40 (72.7%) |
| yes | 15 (27.3%) |
| ulcerative lesion | 2 (3.6%) |
| polypoid lesion | 12 (21.8%) |
| tiny erosions or hematins | 1 (1.8%) |
| Small bowel cleanliness | |
| very poor | 1 (1.8%) |
| poor | 3 (5.5%) |
| fair | 31 (56.4%) |
| good | 17 (30.9%) |
| excellent | 3 (5.5%) |

visualization, and 2.7% (12/440) had poor visualization. The 12 structures with poor mucosal visualization consisted of 1 case of EG junction, 2 cases of cardia & fundus, 5 cases of body, and 4 cases of duodenal bulb.

## The continuous small bowel examination using 3D MACE

After 3D MACE for upper GI examination, the continuous small bowel examination was carried out. The continuous small bowel examination by 3D MACE was completed in 94.5%. The mean small bowel transit time was 260.34 ± 101.14 minutes. Duodenal and jejunal ulcers were seen in two cases each. Small bowel cleanliness was also adequate in 92.8%. (Table 3).

## Safety and patient satisfaction

No adverse events, aspiration or severe pain occurred in the 3D MACE group during upper GI examination. There was no capsule retention or significant adverse events during a follow-up hospital visit at two weeks. However, mucosal injury and minimal bleeding associated with belching occurred in five patients during upper endoscopy. Bleeding was minor and spontaneously stopped without further management. For overall patient satisfaction on a 10-point scale (the higher the score, the higher the satisfaction), 3D MACE scored 9.55 ± 0.79 points and upper endoscopy scored 7.75 ± 2.34, showing higher satisfaction with 3D MACE (p < 0.0001). 96.4% (53 cases) reported no discomfort related to the 3D MACE, and only 3.6% (2 cases) complained of mild discomfort related to the magnetic controller. Regarding discomfort other than the 3D MACE procedure, 34 patients (61.8%) reported discomfort related to taking a PEG purgative, while 5, 2, and 2 patients reported discomfort related to water intake and returning data sensor belt with receiver, respectively. (S2 Table). If the upper GI examination was performed again, the intention to perform 3D MACE was confirmed in 94.6% (52 cases).

## Discussion

In this study, we employed a novel 3D MACE system that provided better image quality than the conventional MACE and allowed for 3D image reconstruction. We evaluated the utility of

a 3D MACE with a hand-held controller for simultaneous examination of the upper GI tract and small bowel. This study results demonstrated a good performance and safety of the 3D MACE system for upper GI and small bowel examination, and supporting its the potential clinical use. The 3D MACE system was able to confirm major gastric structures in 98.6% of cases, with only one exception in which the EG junction was not identified. In addition, the 3D MACE system showed a high accuracy in identifying abnormal gastric lesions. It allowed for a stereoscopic identification of gastric lesions through 3D image reconstruction. In the small bowel examination performed after the gastric examination with 3D MACE, 94.5% of cases were confirmed as complete small bowel examination. Significant adverse events due to the 3D MACE system were not observed. Only two patients complained of discomfort related to the magnetic controller.

The 3D MACE system may be an alternative for patients who cannot tolerate upper endoscopy. 3D MACE does not require sedation and it is a non-invasive endoscopy that allows direct inspection of the GI mucosa. Gastric cancer is the fifth most common type of cancer and the fourth leading cause of cancer-related deaths globally [14]. Especially, gastric cancer is most common in East Asia, Eastern Europe, and South America [15]. Therefore, gastric cancer screening is in progress in Korea and Japan using upper endoscopy and upper GI series (UGIS). However, unlike upper endoscopy, mortality reduction from gastric cancer has not been demonstrated in UGIS [16, 17]. Several studies have also shown superiority of upper endoscopy over UGIS in gastric cancer screening [18, 19]. However, upper endoscopy is an invasive procedure that can cause pain and adverse events to patients [20]. Sedative is administered to reduce endoscopy-related discomfort, but sedation-associated adverse events such as respiratory and cardiac problems can occur in 0.3% [21]. A study comparing conventional MACE and upper endoscopy has reported that MACE can significantly reduce endoscopy-related pain and discomfort [6]. Furthermore, a recent study involving 3,182 participants has found that magnetically controlled capsule gastroscopy (MCCG) with a robotic controller system had similar accuracy as upper endoscopy in detecting gastric cancer [22]. 3D MACE combines benefits of UGIS and upper endoscopy. Thus, this can also be used for gastric cancer screening.

In addition, because 3D MACE is disposable, there is no concern about endoscopy- mediated infections or reprocessing. Although endoscopy-mediated infections and transmission are not common, several cases of carbapenem-resistant Enterobacteriaceae transmission through a duodenoscopy have been reported worldwide [23]. To prevent endoscopy-mediated infection, strict reprocessing is being carried out. However, it also consumes additional resources such as manpower and costs. After the coronavirus disease 2019 (COVID-19) pandemic, there are restrictions on the implementation of endoscopy due to concerns about the spread of COVID-19 infection [24]. However, since 3D MACE is used only once without generating aerosol, it can be free from endoscopy mediated infections and disease transmission.

3D MACE allows for examination of both the upper GI and small bowel simultaneously. Several studies have been conducted using capsule endoscopy equipped with magnets and different types of magnetic controllers for examining the stomach and small bowel [25, 26]. Especially, magnetically controlled capsule endoscopy (MCE) with a robotic controlled system showed good performances to examine both the stomach and small bowel [27–29]. We examined the upper GI and small bowel simultaneously for the first time with a portable 3D MACE system, and also found a good performance like a robotic controlled system. With an increase in battery power and technological advancements of 3D MACE, it will become possible to perform "pan-endoscopy", which would enable examination of the entire GI tract from the mouth to the anus [30].

However, upper GI tract examination using the 3D MACE system had several limitations. First, there were a few cases where the esophageal mucosa could not be confirmed with a 3D MACE. It was difficult to observe the esophageal mucosa due to noise and black-out images. It was thought that data reception might not work well because it was far from the data sensor belt of 3D MACE and the esophagus. A study on the diagnosis of Barrett's esophagus and esophageal varices using MACE was performed with a sensing pad attached to the chest area [8]. It is necessary to confirm whether complete esophageal examination can be performed by improving the data sensor belt. Second, there were also a small number of cases where the gastric mucosa was not well visible. MACE or MCE cannot perform active cleansing during a gastric examination. It was also possible to remove air bubbles accumulated in the fundus and cardia by drinking water or changing posture [31]. However, poor mucosal visualization caused by bile acids or dissolved drugs made it difficult to examine. Unlike upper endoscopy, 3D MACE also has a disadvantage in that the gastric lumen cannot be expanded through air insufflation. Of course, it is possible to expand the gastric lumen to some extent by drinking water during the examination. However, there were cases where the stomach shrank after a certain period of time. Third, the characteristics of the inflamed lesion could not be clearly described because it was not relevant to the outcome of this study. Also, there may be inter-observer differences between endoscopists whether an inflamed lesion was edematous or erosive lesion, and this distinction is thought to be minor. However, the extent of the inflamed lesion could be confirmed similarly for both 3D MACE and upper endoscopy. Forth, no gastric cancer cases were observed in this study. Because cancer detection is one of the most important purposes of endoscopy, additional research on gastric cancer detection using 3D MACE is absolutely necessary. Finally, all MACE, MCE, and MCCG systems cannot perform biopsy of significant gastric lesions. Gastric examination using a magnetic capsule is still in the challenging stage. Therefore, it is considered premature to separate 3D MACE from upper endoscopy. The need for biopsy in 3D MACE is also considered to be the same as the indications for upper endoscopy such as gastric ulcer and suspected gastric cancer. If necessary, upper endoscopy can only be performed again.

Despite these limitations, this study shows that a novel 3D MACE system is capable of simultaneous and complete upper GI and small bowel examination. 3D MACE was as accurate as upper endoscopy for identifying structures and lesions in the upper GI tract. In addition, compared to conventional MACE, the 3D MACE showed image quality improvement and 3D images reconstruction. For individuals who cannot tolerate upper endoscopy, the 3D MACE device may be an option. Further prospective large-scale clinical studies are needed to confirm whether 3D MACE is effective for gastric cancer screening and pan-endoscopy in near future.

## Supporting information

**S1 Fig. Control of three dimensional magnetically assisted capsule endoscopy (3D MACE) direction.** The camera direction of the 3D MACE could be adjusted using the direction of N and S poles on the handle of the controller.
(TIF)

**S2 Fig. The real process of upper gastrointestinal examination using three dimensional magnetically assisted capsule endoscopy (3D MACE) system.**
(TIF)

**S3 Fig. Upper gastrointestinal lesions detected by three dimensional magnetically assisted capsule endoscopy (3D MACE) system.**
(TIF)

**S1 Table. Baseline characteristics of study populations.**
(DOCX)

**S2 Table. Patient satisfaction during examination.**
(DOCX)

**S1 Dataset.**
(ZIP)

**S1 Checklist. TREND statement checklist.**
(PDF)

## Acknowledgments

The authors are thankful to the Intromedic company for developing the 3D MACE system. The company had no research or financial involvement in this study. Researchers of All-Live healthcare company were very helpful in conducting this clinical study.

## Author Contributions

**Conceptualization:** Dong Jun Oh, Yun Jeong Lim.

**Data curation:** Yun Jeong Lim.

**Formal analysis:** Dong Jun Oh.

**Funding acquisition:** Yun Jeong Lim.

**Investigation:** Dong Jun Oh.

**Methodology:** Yun Jeong Lim.

**Resources:** Yun Jeong Lim.

**Supervision:** Hyun Seok Lee, Ji Hyung Nam, Yun Jeong Lim.

**Validation:** Sang Hoon Kim, Joowon Chung, Hyun Seok Lee.

**Writing – original draft:** Dong Jun Oh, Yea Je Lee.

**Writing – review & editing:** Sang Hoon Kim, Joowon Chung, Ji Hyung Nam, Yun Jeong Lim.

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
