## [Decision Letter · Decision Letter 0]

4 Sep 2023

PONE-D-23-18486Efficacy and safety of three-dimensional magnetically assisted capsule endoscopy for upper gastrointestinal and small bowel examinationPLOS ONE

Dear Dr. Lim,

Thank you for submitting your manuscript to PLOS ONE. After careful consideration, we feel that it has merit but does not fully meet PLOS ONE’s publication criteria as it currently stands. Therefore, we invite you to submit a revised version of the manuscript that addresses the points raised during the review process.

We look forward to receiving your revised manuscript.

Kind regards,

Thomas Lui Ka Luen

Academic Editor

PLOS ONE

“This study was supported by a grant (grant Number: HI19C0665) from the Korean Health Technology R & D project through the Korean Health Industry Development Institute (KHIDI) funded by the Ministry of Health & Welfare, Republic of Korea.”

Please respond by return e-mail so that we can amend your financial disclosure and competing interests on your behalf.

4. We note that the original protocol file you uploaded contains a confidentiality notice indicating that the protocol may not be shared publicly or be published. Please note, however, that the PLOS Editorial Policy requires that the original protocol be published alongside your manuscript in the event of acceptance. Please note that should your paper be accepted, all content including the protocol will be published under the Creative Commons Attribution (CC BY) 4.0 license, which means that it will be freely available online, and any third party is permitted to access, download, copy, distribute, and use these materials in any way, even commercially, with proper attribution.

Therefore, we ask that you please seek permission from the study sponsor or body imposing the restriction on sharing this document to publish this protocol under CC BY 4.0 if your work is accepted. We kindly ask that you upload a formal statement signed by an institutional representative clarifying whether you will be able to comply with this policy. Additionally, please upload a clean copy of the protocol with the confidentiality notice (and any copyrighted institutional logos or signatures) removed.

5. We note that the original protocol that you have uploaded as a Supporting Information file contains an institutional logo. As this logo is likely copyrighted, we ask that you please remove it from this file and upload an updated version upon resubmission.

6. We note that Figures 2, 3, 4, 5, S1, S2 and S4 in your submission contain copyrighted images. All PLOS content is published under the Creative Commons Attribution License (CC BY 4.0), which means that the manuscript, images, and Supporting Information files will be freely available online, and any third party is permitted to access, download, copy, distribute, and use these materials in any way, even commercially, with proper attribution. For more information, see our copyright guidelines: http://journals.plos.org/plosone/s/licenses-and-copyright.

1. You may seek permission from the original copyright holder of Figures 2, 3, 4, 5, S1, S2 and S4 to publish the content specifically under the CC BY 4.0 license.

Reviewers' comments:

Reviewer's Responses to Questions

**Comments to the Author**

1. Is the manuscript technically sound, and do the data support the conclusions?

Reviewer #1: Yes

Reviewer #2: Partly

2. Has the statistical analysis been performed appropriately and rigorously? 

Reviewer #1: Yes

Reviewer #2: Yes

3. Have the authors made all data underlying the findings in their manuscript fully available?

Reviewer #1: Yes

Reviewer #2: No

4. Is the manuscript presented in an intelligible fashion and written in standard English?

Reviewer #1: Yes

Reviewer #2: Yes

5. Review Comments to the Author

Reviewer #1: This is a well-designed study to show the potential application of 3D MACE in evaluating upper GI and small bowel lesions. The strong point is that to have the endoscopy result to compare. The similarity in results suggested further application of this technique in the future for the patients not tolerating to endoscopy.

However, due to the small number of patients, the data analysis is simple with descriptive methods mainly. I have some suggestions for the authors:

- Clarify more in the results besides detection, is it possible for the doctors to evaluate the characteristics of the lesions using the standardized classification? For the inflammatory lesions, can 3D MACE detect both the lesions with focal and spreading areas? If possible adding more characteristics of the lesions and endoscopists' evaluations will make the data more interesting.

- For the cases with mucosal injury and minor bleeding, in fact what is the origin of the bleeding? EGD is safe so it is a bit confusing why patients have post-procedure bleeding?

Reviewer #2: Congratulation to Lim et al on their proof of concept study utilising a novel 3D enhanced MACE for examination of stomach and small bowel in 55 patients. The efficacy and safety appears to be comparable with conventional upper endoscopy. My comments on the manuscript are listed as following:

Financial disclosure

- Intromedic is mentioned as clinical sponsor in the study protocol which is not stated in the financial disclosure. Please clarify Intromedic's role in the study

Method

- Further description on manipulation of the MACE after ingestion should be provided.

- Please define unsuccessful examination of anatomical structure or suspected lesions eg. length of time or number of attempts in assessment of a structure or lesion of concern

Result and discussion

- Photos taken by MACE vs. 3D MACE vs. conventional endoscopy on selected case should be provided for better appreciation by the readers

- For figure 5 regarding lesions detected, please comment on whether further assessment of lesions ie. image magnification, image enhancement is possible. Also, how to decide on need for conventional upper endoscopy with biopsy if the 3D-MACE is to be used independently.

- Image on small bowel examination including detected pathology should be provided

- Regarding the 3 inflamed lesions and 1 polyps detected by 3D MACE but not upper endoscopy, how are the cases managed ? Are they considered as false positive or follow up assessment is arranged to confirm the lesion ?

- Additional limitation of the study is no gastric cancer cases being recruited, therefore efficacy in detecting potential malignant lesion cannot be inferred

6. PLOS authors have the option to publish the peer review history of their article (what does this mean?). If published, this will include your full peer review and any attached files.

Reviewer #1: **Yes: **Dao Viet Hang

Reviewer #2: No

---

## [Author Response · Author response to Decision Letter 0]

6 Oct 2023

Thank you very much for giving the opportunity of revision. The authors appreciate the reviewer’s critical and thoughtful comments. We did our best to write the reply for your comments. We also used the "Track Changes" feature to make revision to the manuscript and make them easily visible to editors and reviewers.

[Reviewer 1]

1. Clarify more in the results besides detection, is it possible for the doctors to evaluate the characteristics of the lesions using the standardized classification? For the inflammatory lesions, can 3D MACE detect both the lesions with focal and spreading areas? If possible adding more characteristics of the lesions and endoscopists' evaluations will make the data more interesting.

: This is an important point. Of course, 3D MACE operator and endoscopist each classified the characteristics of the inflamed lesion such as erosive, hyperemic, and edematous. However, subgroup analysis was not performed because there were differences between 3D MACE operator and endoscopist in distinguishing whether an inflamed lesion was hyperemic or erosive. Additionally, this distinction was thought to be minor and insignificant, and was not relevant to the outcome of this study.

: The extent of the inflamed lesion could be confirmed similarly for both 3D MACE and upper endoscopy. This information added to the “Result; Table 2” and “Discussion; Limitations” section.

2. For the cases with mucosal injury and minor bleeding, in fact what is the origin of the bleeding? EGD is safe so it is a bit confusing why patients have post-procedure bleeding?

: Most procedure related bleeding involved belching in this study. The bleeding was spontaneously stopped without any additional procedure. We added this content to the “Result; Safety and patient satisfaction” section.

[Reviewer 2]

1. Financial disclosure - Intromedic is mentioned as clinical sponsor in the study protocol which is not stated in the financial disclosure. Please clarify Intromedic's role in the study

: Intromedic company only developed 3D MACE for a research project and we did the research independently. The company had no research or financial involvement in this study. Additionally, the authors have no financial or personal relationship with Intromedic. We added this content in the “Acknowledgments” section.

2. Method - Further description on manipulation of the MACE after ingestion should be provided.

: We added a description of the manipulation of 3D MACE in “Method; Study protocol; 3D MACE for upper GI and small bowel examination” section

→ “When the 3D MACE stopped at the esophagus, the controller was moved up and down to observe the esophagus.” “Once the 3D MACE landed on the stomach, the cardia and fundus were observed by rotating and tilting the controller in the left lateral position. Then, the patient was placed upright, and the controller was placed in the left upper quadrant and left flank area to observe the fundus and body. After that, the operator moved the controller to right upper quadrant and right flank area to observe the antrum and pylorus. If necessary, the patient was changed to the right lateral position and the antrum was observed.”

3. Method - Please define unsuccessful examination of anatomical structure or suspected lesions eg. length of time or number of attempts in assessment of a structure or lesion of concern

: This is a good point. Mucosal visualization for each structure was also assessed. Failure to structure confirmation was defined as a case where the structure was not visible during examination or was not clearly photographed. There were no restrictions on length of time or number of attempts. We added this content in the “Method; Outcomes and Safety” section.

4. Result and discussion - Photos taken by MACE vs. 3D MACE vs. conventional endoscopy on selected case should be provided for better appreciation by the readers

: MACE and 3D MACE are completely different models. And since this study was approved for clinical research only using 3D MACE, there is no data regarding MACE. Therefore, MACE and 3D MACE could not be compared in this study. Instead, we will post here images of MACE published in other articles, for comparison with 3D MACE.

5. Result and discussion - For figure 5 regarding lesions detected, please comment on whether further assessment of lesions ie. image magnification, image enhancement is possible. Also, how to decide on need for conventional upper endoscopy with biopsy if the 3D-MACE is to be used independently.

: Image brightness and definition was enhanced using a reading viewer. We added this content in the “Figure 5” section. Gastric examination using a magnetic capsule is still in the challenging stage. Therefore, it is considered premature to separate 3D MACE from upper endoscopy. The need for biopsy in 3D MACE is also considered to be the same as the indications for upper endoscopy such as gastric ulcer and suspected gastric cancer. We added this content in the “Discussion; Limitation” section.

6. Result and discussion - Image on small bowel examination including detected pathology should be provided

: Images of small bowel ulcers were included in S4 Fig.

7. Result and discussion - Regarding the 3 inflamed lesions and 1 polyps detected by 3D MACE but not upper endoscopy, how are the cases managed? Are they considered as false positive or follow up assessment is arranged to confirm the lesion?

: Thank you for your thoughtful comment. In the case of missed lesions, the authors reviewed the 3D MACE images and concluded that the lesions were insignificant. The endoscopist explained that the patients needed periodic follow-up endoscopy. We added this information in the “Result” section.

8. Result and discussion - Additional limitation of the study is no gastric cancer cases being recruited, therefore efficacy in detecting potential malignant lesion cannot be inferred

: I agree with your comment. We added to the “Discussion” section the limitation that we could not confirm the efficacy of gastric cancer detection because we did not have gastric cancer patients.

---

## [Decision Letter · Decision Letter 1]

13 Nov 2023

PONE-D-23-18486R1Efficacy and safety of three-dimensional magnetically assisted capsule endoscopy for upper gastrointestinal and small bowel examinationPLOS ONE

Dear Dr. Lim,

Thank you for submitting your manuscript to PLOS ONE. After careful consideration, we feel that it has merit but does not fully meet PLOS ONE’s publication criteria as it currently stands. Therefore, we invite you to submit a revised version of the manuscript that addresses the points raised during the review process.

We look forward to receiving your revised manuscript.

Kind regards,

Thomas Lui Ka Luen

Academic Editor

PLOS ONE

Journal Requirements:

Reviewers' comments:

Reviewer's Responses to Questions

**Comments to the Author**

1. If the authors have adequately addressed your comments raised in a previous round of review and you feel that this manuscript is now acceptable for publication, you may indicate that here to bypass the “Comments to the Author” section, enter your conflict of interest statement in the “Confidential to Editor” section, and submit your "Accept" recommendation.

Reviewer #3: (No Response)

2. Is the manuscript technically sound, and do the data support the conclusions?

Reviewer #3: Yes

3. Has the statistical analysis been performed appropriately and rigorously? 

Reviewer #3: Yes

4. Have the authors made all data underlying the findings in their manuscript fully available?

Reviewer #3: Yes

5. Is the manuscript presented in an intelligible fashion and written in standard English?

Reviewer #3: Yes

6. Review Comments to the Author

Reviewer #3: On the overall, the authors were able to address the previous round of comments. Some inconsistencies are still present in the statistical section:

(a) The power/sample size stab does not mention the statistical test that was used.

(b) It is not clear what the authors meant by "Kapp" in "Kapp, sensitivity, and specificity were used to assess..."

7. PLOS authors have the option to publish the peer review history of their article (what does this mean?). If published, this will include your full peer review and any attached files.

Reviewer #3: No

---

## [Author Response · Author response to Decision Letter 1]

26 Nov 2023

Response to reviewer

Thank you very much for giving the opportunity of revision. The authors appreciate the reviewer’s critical and thoughtful comments. We did our best to write the reply for your comments. We also used the "Track Changes" feature to make revision to the manuscript and make them easily visible to editors and reviewer.

On the overall, the authors were able to address the previous round of comments. Some inconsistencies are still present in the statistical section:

1. The power/sample size stab does not mention the statistical test that was used.

: Thank you for the helpful suggestion. We revised the sentence as follows in relation to mention of power/sample size stability: ‘To ensure the power and stability of the sample size in our study, we conducted a Power Analysis, setting the level of significance (α) at 0.05 and the power of the test (β) at 80%. This involved considering a standard deviation of 4.0 based on the successful visualization of gastric structures from the previous study [5], and an effect size of 0.8 in the t-test for the Power Analysis.

2. It is not clear what the authors meant by "Kapp" in "Kapp, sensitivity, and specificity were used to assess..."

: Kapp is a misspelling of kappa (k), and I apologize for the mistake. We revised the sentence as follows: ‘Kappa (κ), sensitivity, and specificity were used to evaluate the agreement and assess each part of upper GI anatomy and gastric lesions between two groups.

---

## [Editor Report · Decision Letter 2]

29 Nov 2023

Efficacy and safety of three-dimensional magnetically assisted capsule endoscopy for upper gastrointestinal and small bowel examination

PONE-D-23-18486R2

Dear Dr. Lim,

We’re pleased to inform you that your manuscript has been judged scientifically suitable for publication and will be formally accepted for publication once it meets all outstanding technical requirements.

Kind regards,

Thomas Lui Ka Luen

Academic Editor

PLOS ONE